# Arbuscular Mycorrhizal Fungi as a Plant Growth Stimulant in a Tomato and Onion Intercropping System

Muhammad Shafiq [1], Josefina Casas-Solís [1,*], Cecilia Neri-Luna [1], Munazza Kiran [2], Saba Yasin [3], Diego Raymundo González-Eguiarte [1] and Alejandro Muñoz-Urias [1]

[1] Centro Universitario de Ciencias Biológicas y Agropecuarias, Universidad de Guadalajara, Zapopan 45110, Mexico; muhammad.shafiq@alumnos.udg.mx (M.S.); cecilia.neri@academicos.udg.mx (C.N.-L.); diego.geguiarte@academicos.udg.mx (D.R.G.-E.); alejandro.munozu@academicos.udg.mx (A.M.-U.)

[2] Department of Botany, Division of Science & Technology, University of Education, Lahore 54770, Pakistan; munazza.kiran@ue.edu.pk

[3] Facultad de Agronomía, Universidad Autónoma de Nuevo León, San Nicolás de los Garza 66455, Mexico; saba.yasinx@uanl.edu.mx

\* Correspondence: josefina.casas@academicos.udg.mx

**Abstract:** Climate change has challenged large-scale crop production at a global level. Global temperature increases, water scarcity, and a further reduction in cultivable land resources due to anthropogenic impacts have resulted in the need to redesign agricultural systems such as intercropping to maximize the efficient use of natural resources. Arbuscular mycorrhizal fungi (AMF) represent an underexplored area, not only in terms of an alternative to the heavy use of chemical fertilizers, but also as a natural resource used to enhance physiological processes and mitigate the variations in biotic and abiotic factors in plants. On the other hand, the combined use of AMF with suitable but cheaper and environmentally friendly growth substrates is another way to maximize crop production. A study was carried out in a tomato and onion intercropped pattern system to analyze the above- and belowground implications of two AMF commercial products containing *Rhizophagus irregularis*, propagated in soil and with an in vitro technique addition, with two different mixed growth substrates (river sand and compost) under greenhouse conditions. Overall, both AMF products overall showed significant promoting effects on plant growth (15–30%) and root parameters (50%) in the tomato and onion plants on the sand-mixed substrate. Moreover, the soil-propagated AMF also showed significant positive effects on chlorophyll content (35%), photosynthetic activity, and the accumulation of macro- and micronutrients, especially the Fe and Mn contents (60–80%) in the tomato plants. We present evidence of the benefits to plant performance due to the interactive effects between AMF and the growth substrate, and these positive effects might be due to the intercropping system. Hence, soil-propagated *Rhizophagus irregularis* is represented here as a promising candidate for enhancing growth, sustainability, and productivity under greenhouse conditions.

**Keywords:** intercropping; *Rhizophagus irregularis*; growth substrate; micronutrients; root architecture

## 1. Introduction

Climate change and the constant exploitation of natural resources have been crucial triggers in the search for alternate and innovative crop production systems. Greenhouses have been helpful in growing off-season vegetables and fruits over the years as they provide crops with a shield against climatic impacts and disasters, ensuring stable crop growth [1]. During the last decade, greenhouse production of high-value vegetables such as tomato and onion under a controlled environment has shown tremendous potential to attain maximum productivity [2]. Tomato and onion are two of the world's most-consumed vegetables, belonging to the Solanaceae and Amaryllidaceae families, respectively [3]. The chief tomato producers are Asia (53.8%) and the Americas (17.9%) [4]. Meanwhile, Mexico is one of the

largest tomato exporters in the world after Holland [5], producing 3780.95 thousand tons of tomato during 2020, of which 37.6% were produced under protected conditions. On the other hand, onion is produced globally as an important vegetable crop [6] for income generation in many developing countries [7]. Global onion production was recorded at more than 100 million tons on a total area of 5.2 million hectares in 2019, making it the third-largest vegetable crop by production weight [8]. Tomato and onion crops can also be grown together via intercropping, which favors the microclimatic conditions around the canopy, adding organic matter and nitrogen to the soil, retaining water and nutrients, and suppressing weeds in greenhouse crop production [9]. It has been reported that choosing a suitable plant species for intercropping significantly reduces the environmental footprint and increases crop yields markedly up to 15–25% compared with monoculture cropping [10–12].

Many factors play critical roles in the successful crop production of tomato and onion, such as traditional fertilization. However, the heavy use of chemical fertilizers and less available fertile land has provoked environmental pollution and future food safety risks in such horticultural crop production [13]. Among suitable practices for the sustainable management of agricultural soils, the use of biostimulants, bioinoculants, and biofertilizers is becoming increasingly common. Arbuscular mycorrhizal fungi (AMF) can lead to successful plant growth, yield enhancement, and improvement in fruit quality [14]. The AMF provide symbiotic associations between plants and soil fungi [15] and belong to the phylum Glomeromycota [16]. One of the identified AMF, *Rhizophagus irregularis* (Błaszk., Wubet, Renker & Buscot) C. Walker & A. Schüßler [17–19] is widely used as a biofertilizer in sustainable crop production to colonize plant roots and promote plant growth and physiology with noticeable effects on shoots and fruit [20]. AMF-colonized roots can promote plant growth and physiology with observable effects on shoots, roots, and fruit quality [21]. Moreover, AMF are known to increase the bioavailability of P in the rhizosphere and significantly enhance the utilization of N by the inoculated plant. The AMF regulate the rhizosphere microenvironment of host plants by secreting substances and influencing the species and proportion of plant-root secretions [22].

Furthermore, suitable growth substrates with different compositions of organic and inert materials play a critical role in the efficient production of tomato and onion, helping them to develop properly, rapidly, and uniformly until they are fully established [23]. Many researchers started looking at the alternatives to peat that were being sought worldwide in the late 1970s [24] by testing the viability of compost, coconut fiber, and volcanic rock as components of substrate mixes for tomato seedlings [25,26]. Coconut fiber and volcanic rocks are byproducts, and using these as a substrate component promotes sustainability and reduces waste. Under different conditions and combinations, mixed or single materials can be used to produce different substrates, keeping in mind the required physical, chemical, and biological characteristics [27]. Sand-mixed substrates are known for their excellent drainage properties due to the coarse texture of sand particles. When combined with coconut fiber and volcanic rock, these substrates help prevent waterlogging, which can lead to root suffocation and the development of root diseases [28]. The loose structure of sand substrates, combined with the fibrous nature of coconut fiber, promotes root aeration, facilitating proper respiration and nutrient uptake by the roots. Coconut fiber, also known as coir, and volcanic rock, when mixed with sand substrates helps maintain a more stable moisture level in the root zone, preventing the excessive drying out of the substrate [29]. Moreover, coconut fiber has a high cation-exchange capacity, enabling it to retain and release essential nutrients to plants gradually. It acts as a reservoir for nutrients, preventing leaching and ensuring their availability to plant roots over time. Combining coconut fiber and volcanic rock with sand substrates helps create a nutrient-rich and well-balanced growing medium for plants [29]. Additionally, the natural properties of coconut fiber contribute to the overall eco-friendliness of the substrate mixture [29,30].

Previously research studies have shown that the application of AMF species on horticultural crops improves agronomic, physiological, and nutritional parameters. To the

best of our knowledge, the current research work creates, for the first time, a sustainable intercropping system using biostimulants such as *R. irregularis* along with eco-friendly growing substrates, and reveals their effects on above- and belowground parameters in tomato and onion under greenhouse conditions. Furthermore, we aim to examine if one of these two mixed substrates is compatible with *R. irregularis* and whether it had an impact on the root architectural parameters.

## 2. Materials and Methods

### 2.1. Plant Material

A pot experiment was conducted at the Center of Biological and Agricultural Sciences, University of Guadalajara, Mexico. Tomato (cv. Josefina) and onion (cv. cebolla blanca) seeds were obtained from Semillas Caloro, Guadalajara, Mexico. Both crop seeds were surface sterilized by gently shaking in a 5% sodium hypochlorite (NaOCl) solution for 35 min and then rinsed with sterile distilled water six times [31]. The seeds were pre-germinated on moist sterile filter paper in Petri dishes and placed in an incubator (NOVATECH DBO-200) at 21 °C.

### 2.2. Experimental Set Up and Crop Management

Eight-day-old tomato and onion plantlets were transplanted into 8 kg plastic pots (21 cm × 27 cm) filled with two different growth substrates. These substrates, based on volcanic rock and coconut fiber mixed either with river sand (SVC) or compost (CVC) in a 2:1:1 proportion, were screened (≈2 mm) and autoclaved (Yamato sterilizer SM300) at 0.10 MPa and 121 °C for 1 h (three times) using the moist heat method [32]. The physicochemical properties of each substrate are shown in Table 1. Two commercial AMF products (provided by Biosustenta®, Morelia, Michoacan, Mexico) were tested: (i) soil-multiplied *R. irregularis* containing spores and root fragments (AMF+) and (ii) in vitro *R. irregularis* (AMF++). AMF+ inoculation was carried out 3 cm below substrate in the shape of bands, while AMF++ was applied as fertigation during transplanting, calculating approximately 250–300 spores per pot. Later, inoculation was repeated three times during different growth stages: 15 (root initiation), 40 (flowering stage), and 60 (fruit-development stage) days after transplanting (DAT).

**Table 1.** Physical and chemical properties of growing substrates (mgL$^{-1}$).

| Substrate | pH | EC (dSm$^{-1}$) | NO$_3^-$ | NH$_4^+$ | PO$_4^{3-}$ | Fe$^{2+}$ | Cu$^{2+}$ | Zn$^{2+}$ | Mn$^{2+}$ |
|---|---|---|---|---|---|---|---|---|---|
| SVC | 7.4 | 0.2 | 5 | 3 | 0.70 | 0.46 | 0.1 | 0.03 | 0.03 |
| CVC | 7.6 | 1.2 | 3 | 5 | 13 | 0.73 | 0.2 | 0.2 | 1.3 |

The plants were transferred to a greenhouse keeping row–row and pot–pot distances at 60 and 40 cm, respectively. The pots were organized in a completely randomized design consisting of six treatments, and each treatment was replicated 10 times. The treatments were arranged as follows: (i) SVC without AMF inoculation (control), (ii) AMF+ inoculation with SVC, (iii) AMF++ inoculation with SVC, (iv) CVC without AMF inoculation (control), (v) AMF+ inoculation with CVC, and (iv) AMF++ inoculation with CVC. Later, the pots were maintained in the greenhouse under a day/night temperature of 25.3–32.5 °C, relative humidity between 67% and 83%, and a photoperiod of 13/11 h. All plants were watered daily with tap water for 20 min using drip irrigation (5 L h$^{-1}$), and once a week with a modified Steiner nutrient solution at each plant growth stage [33].

### 2.3. Root Colonization (%)

Roots (0.5 g) were carefully washed and cut into 1 cm segments. Root segments were cleared with 10% potassium hydroxide (KOH) solution at 90 °C for 10 min and rinsed with water prior to acidification with M/10 HCl overnight. The root segments were then stained with 0.05% trypan blue [34]. The mycorrhizal colonization rate was measured at 65 DAT

using the magnified intersection method [35]. The percentage of roots colonized by AMF was calculated as follows:

$$root\ colonization = \frac{number\ of\ colonized\ root\ sections}{total\ number\ of\ root\ sections} \times 100\%$$

### 2.4. Root Morphology

Root samples of five plants from each treatment of both tomato and onion were taken 65 DAT and carefully washed. The excised root system of each plant was digitized using the WinRhizo LA2400 scanner (Regent instruments®, Québec, CA, Canada). The analysis of the digitized root images was performed using WinRhizo TM Pro 2016a software (2016) at a scanning resolution of 300 dpi. To describe the root architecture, the following morphometric variables were measured: (a) root length (cm), (b) diameter (cm), (c) surface area (cm$^2$), (d) volume (cm$^3$), (e) number of forks, (f) crossings, and (g) root tips [36].

### 2.5. Morpho-Agronomic Traits

Tomato and onion plant agronomic parameters were evaluated by selecting five random plants per treatment and measuring the plant height above the substrate level (2 cm) and the stem diameter using a digital Vernier (Steren HER-411, Steren Electronics International LLC, San Diego, CA, USA). The total number of flowers per plant from five randomly selected tomato plants was recorded 40 DAT. Five onion plants per treatment were selected to evaluate the total number of leaves at 60 DAT.

### 2.6. Chlorophyll Contents

Chlorophyll extraction of tomato and onion was carried out according to Bruinsma's method (1983) [37] at 60 DAT by grinding fresh leaves (2 g) in 10 mL of acetone (80%), followed by centrifugation of homogenate ($10,000 \times g$; Hermle Z300) for 10 min. Chlorophyll a, chlorophyll b, and total chlorophyll content was determined at absorbances of 645 and 663 nm using the spectrophotometric (VELAB mod VE-6000T) method.

### 2.7. Leaf-Mineral Contents

Three fully expanded tomato leaves were collected from all experimental treatments (60 DAT) to estimate the effect of the AMF inoculum and substrate levels on the nutrient status. The leaves were carefully washed with $H_2O$ and 4 N HCl solution, rinsed with distilled water, oven-dried at 60 °C for 2 days, ground using a mortar, and passed through a 2 mm mesh sieve. Subsequently, 2 g of dried powdered material was obtained from each treatment. Nitrogen (N) was quantified using the Kjeldahl method (Hanon K9840 Kjeldahl apparatus), as described by Yan et al., 2012 [38], and total carbon (Ct) was measured according to Kalra, 1997 [39]. Iron (Fe), copper (Cu), zinc (Zn), and manganese (Mn) were evaluated using the digestion method. For P, the UV–visible spectrophotometric technique was used [40].

### 2.8. Photosynthetic Activities

The net photosynthetic rate (Pn), transpiration rate (E), $CO_2$ internal concentration (Ci), and stomatal conductance (gs) in the tomato plants were measured 60 DAT using a portable photosynthetic system (Li-6400, PSC 1376, Inc., Lincoln, NE, USA) from 10:00 to 12:00 h under natural sunlight. The air temperature was 23 °C and the external $CO_2$ concentration was 300–350 ppm. The third pinnate compound leaf below the growing point was selected from five random plants in each treatment group [41]. When measuring, the leaf was clamped in a standard leaf chamber of 3–2 cm, with the leaf surface facing up, and the leaf chamber was flat. When the concentration of $CO_2$ in the sample chamber was stable, the data were recorded [42].

### 2.9. Statistical Analysis

The normality of the replicated data was tested using the Shapiro–Wilk test. The data that followed a normal distribution were compiled for further two-way analysis of variance (ANOVA) by assuming the AMF as the first factor and the substrate as the second factor using SPSS (IBM SPSS 22.0, IBM Corporation, New York, NY, USA). The significance of the differences among treatments with regard to various factors were calculated at 5%, and means were compared through the Tukey test ($\alpha = 0.05$). To examine the data of growth parameters, the root architecture and physiological parameters were based on five replicates, whereas the determination of other parameters (mineral contents) was performed using destructive methods based on three replicates.

## 3. Results

### 3.1. Root Colonization (%)

The tomato and onion plants grown together and inoculated with *R. irregularis* on two different growth substrates showed average root colonization of 44% (AMF+) and 36% (AMF++) on the SVC substrate, while the tomato plants revealed rates of 33% (AMF+) and 29% (AMF++) on the CVC substrate. AMF inoculation in the onion plants showed root colonization of 48% (AMF+) and 46% (AMF++) on the SVC substrate and 35% (AMF+) and 37% (AMF++) on the CVC substrate. No colonization was observed in non-inoculated tomato and onion plant roots on either type of growth substrate.

### 3.2. Effects on Root Morphology

Photos of the root system showed that the non-inoculated tomato and onion plants in both substrates had a lesser root development. The application of *R. irregularis* from two different sources of propagation had a significant influence on the root architecture of the intercropped tomato and onion system using a combination of two substrates (Figure 1). The tomato and onion roots showed a significant increase in length (11%, 10%), diameter (63%, 66%), surface area (45%, 26%), volume (44%, 55%), forks (27%, 39%), crossings (24%, 18%), and tips (21%, 68%) when using AMF+ in the SVC substrate compared with the non-inoculated plants (Tables 2 and 3). Similar effects of AMF++ inoculation were observed in the CVC substrate except for RV, RC, and RT in tomato and RSA, RV, and RC in onion roots. Two-way ANOVA showed highly significant interaction effects of AMF inoculation and growth substrate on RF and RC, and a significant difference on RT in tomato. Meanwhile, all parameters except RSA showed significant interaction between AMF inoculation and growth substrate in onion.

**Table 2.** Effect of two commercial products of *R. irregularis* on root architecture parameters in tomato plants grown on two mixed substrates.

| | Treatments | RL (cm) | RD (cm) | RSA (cm²) | RV (cm³) | RF | RC | RT |
|---|---|---|---|---|---|---|---|---|
| SVC | Control | 5132 ± 65 b | 0.66 ± 0.01 c | 993 ± 18 c | 30 ± 1.7 b | 34,463 ± 152 c | 3620 ± 93 b | 7049 ± 131 b |
| | AMF+ | 5691 ± 73 a | 1.08 ± 0.04 a | 1440 ± 22 a | 44 ± 1.6 a | 43,923 ± 124 a | 4509 ± 93 a | 8578 ± 116 a |
| | AMF++ | 5483 ± 59 a | 0.83 ± 0.01 b | 1223 ± 33 b | 33 ± 1.7 b | 36,005 ± 109 b | 3752 ± 96 b | 7538 ± 98 b |
| CVC | Control | 5013 ± 69 b | 0.81 ± 0.01 b | 1123 ± 14 c | 28 ± 2.1 b | 35,455 ± 199 c | 3635 ± 87 a | 7255 ± 92 a |
| | AMF+ | 5467 ± 48 a | 1.10 ± 0.04 a | 1284 ± 12 a | 38 ± 1.4 a | 38,375 ± 178 b | 3945 ± 108 b | 7275 ± 135 a |
| | AMF++ | 5318 ± 40 a | 1.02 ± 0.03 a | 1207 ± 18 b | 33 ± 2.0 b | 40,404 ± 174 a | 3834 ± 131 a | 7420 ± 107 a |
| | AMF | <0.05 | <0.001 | <0.001 | <0.05 | <0.001 | <0.001 | <0.001 |
| | S | ns | <0.05 | <0.05 | ns | ns | <0.01 | <0.05 |
| | AMF × S | ns | ns | ns | ns | <0.001 | <0.001 | <0.05 |

Means within a column followed by the same letter(s) are not significantly different (ns) at *p* > 0.05 based on the Tukey test. The *p*-value indicates a significance level based on a two-way ANOVA. RL = root length, RD = average diameter, RSA = surface area, RV = volume, RF = forks, RC = crossings, and RT = tips. AMF+, soil-multiplied *R. irregularis*; AMF++, in vitro *R. irregularis*; AMF, arbuscular mycorrhizal fungi; S, substrate; AMF × S, indicates interaction between AMF and substrate; CVC, compost; SVC, river Sand.



**Table 3.** Effect of two commercial products of *R. irregularis* on root architecture parameters in onion plants grown on two mixed substrates.

| | Treatments | RL (cm) | RD (cm) | RSA (cm$^2$) | RV (cm$^3$) | RF | RC | RT |
|---|---|---|---|---|---|---|---|---|
| SVC | Control | 636 ± 1.2 b | 0.7 ± 0.09 c | 269 ± 1.17 c | 2.8 ± 0.03 b | 3173 ± 0.93 c | 925 ± 1.14 b | 111 ± 0.81 b |
| | AMF+ | 1510 ± 1.2 a | 1.2 ± 0.05 a | 342 ± 0.98 a | 4.4 ± 0.04 a | 4438 ± 1.04 a | 1099 ± 0.77 a | 187 ± 1.12 a |
| | AMF++ | 1528 ± 1.5 a | 1.1 ± 0.05 b | 308 ± 1.28 b | 4.1 ± 0.05 a | 4249 ± 0.97 b | 1107 ± 0.78 a | 169 ± 1.04 a |
| CVC | Control | 671 ± 1.0 c | 0.86 ± 0.04 b | 256 ± 1.14 a | 3.0 ± 0.05 a | 3325 ± 0.93 c | 980 ± 0.87 a | 120 ± 0.93 b |
| | AMF+ | 1241 ± 1.7 b | 1.07 ± 0.06 a | 264 ± 1.96 a | 3.5 ± 0.05 a | 3966 ± 0.86 b | 997 ± 1.18 a | 159 ± 0.96 a |
| | AMF++ | 1316 ± 1.0 a | 1.1 ± 0.06 a | 276 ± 2.28 a | 3.5 ± 0.04 a | 4137 ± 0.85 a | 1022 ± 0.82 a | 141 ± 1.00 a |
| | AMF | <0.001 | <0.001 | <0.05 | <0.001 | <0.001 | <0.001 | <0.05 |
| | S | <0.001 | ns | <0.01 | <0.001 | <0.001 | <0.05 | <0.001 |
| | AMF × S | <0.001 | <0.001 | ns | <0.001 | <0.001 | <0.01 | <0.001 |

Means within a column followed by the same letter(s) are not significantly different (ns) at $p > 0.05$ based on the Tukey test. The *p*-value indicates a significance level based on a two-way ANOVA. RL = root length, RD = average diameter, RSA = surface area, RV = volume, RF = forks, RC = crossings, and RT = tips. AMF+, soil-multiplied *R. irregularis*; AMF++, in vitro *R. irregularis*; AMF, arbuscular mycorrhizal fungi; S, substrate; AMF × S, indicates interaction between AMF and substrate; CVC, compost; SVC, river Sand.

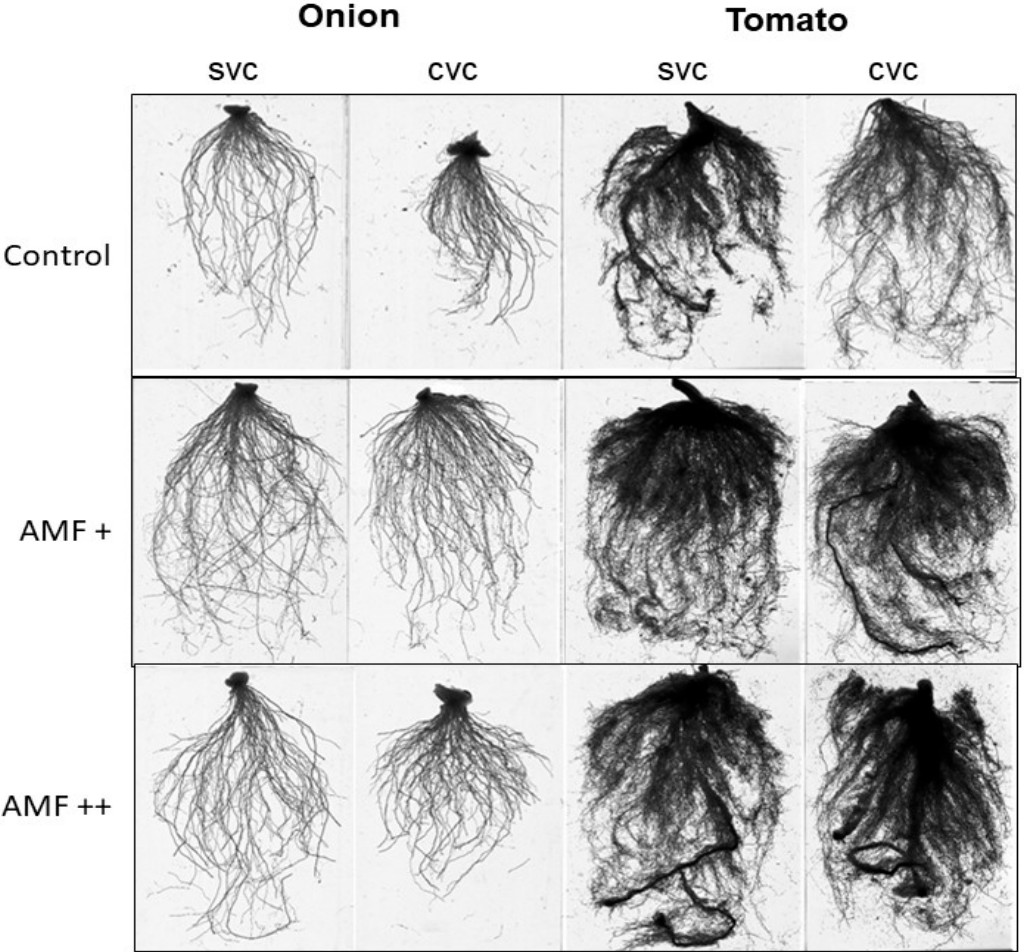

**Figure 1.** Effect of different AMF products on root system architecture in onion and tomato plants grown on two mixed substrates. AMF+, soil-multiplied *R. irregularis*; AMF++, in vitro *R. irregularis*; SVC, river sand: volcanic rock: coconut fiber; CVC, compost: volcanic rock: coconut fiber.

### 3.3. Effect of AMF on Morpho-Agronomic Parameters

In this study, both AMF+ and AMF++ inoculation resulted in a significant difference in most of the measured agronomic parameters of both crops grown in the SVC substrate compared with non-inoculated plants. For instance, AMF+ inoculation in the SVC substrate resulted in an increase of 25% and 43% in tomato plant height and stem diameter,

respectively; meanwhile, AMF++ inoculation resulted in an increase of 19% and 36%, respectively, compared with the control treatments (Figure 2a,b). The number of flowers in the tomato plants equally increased by 50% with AMF+ and AMF++ inoculation compared with the non-inoculated plants (Figure 2c).

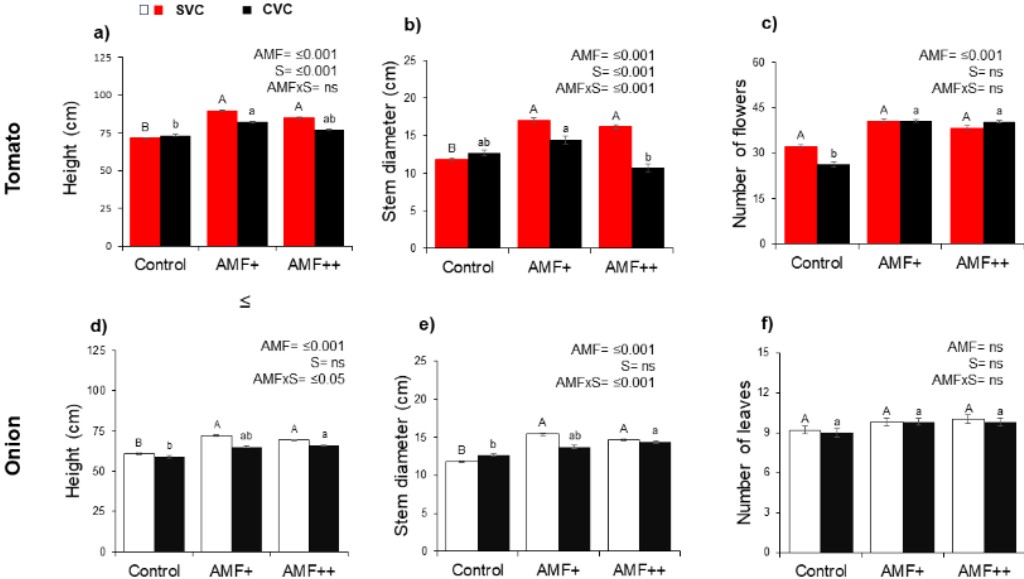

**Figure 2.** Effect of AMF inoculation on the morpho agronomic traits of tomato (**a–c**) and onion (**d–f**) plants grown in two different mixed substrates. Vertical bars show $\pm$ standard errors for mean (*n* = 5). Different letters (capital letters for SVC and lowercase letters for CVC) indicate significant differences according to the Tukey test (*p* < 0.05), and the *p*-value indicates a significance level based on two-way ANOVA.

AMF+ inoculation significantly increased the onion plant height (18%) and stem diameter (30%) in the SVC substrate. However, the plants inoculated with AMF++ in the SVC and CVC substrates showed increases of 14% and 12%, respectively, in plant height and increases of 24% and 14%, respectively, in stem diameter. No significant difference in the number of leaves was found in the onion plants. Among all the morpho agronomic traits, the stem diameter of both crops was significantly affected by the interaction between AMF and the substrate (Figure 2d–f).

### 3.4. Chlorophyll Contents

Total chlorophyll in the tomato leaves was significantly increased by 89% and 54%, while Chl a increased by 79% and 72% with AMF+ and AMF++, respectively, using the SVC substrate (Figure 3a,b), compared with the non-inoculated plants. Furthermore, Chl b content was doubled with AMF+ in the SVC substrate (Figure 3c). The application of two different inoculums in the tomato plants showed similar effects in terms of chlorophyll content in the CVC substrate. In contrast, the Chl total and the Chl a and Chl b contents in the onion plants (Figure 3d–f) did not show a significant increase under both AMF products on the sand- or compost-mixed substrate. Finally, the two-way ANOVA revealed significant interaction between AMF and the growth substrate only in the Chl a pigment of the tomato leaves.

### 3.5. Leaf Mineral Concentration

The AMF inoculum significantly favored the macronutrient contents of N and P in the tomato leaves with SVC compared with the non-inoculated plants. However, the Ct did not a show significant difference under any of the AMF inoculum. In the case of micronutrients, a significant increase in Fe and Mn of more than two times was determined, followed by Cu and Zn, with both AMF inoculations with SVC (Table 4) compared with the non-

inoculated plants. Therefore, the micronutrient concentrations in the tomato leaves were highly stimulated by *R. irregularis* inoculation along with the SVC substrate. The combined analysis of variance revealed the significant effect of AMF and the growth substrate on N, Fe, Mn, and Zn contents.

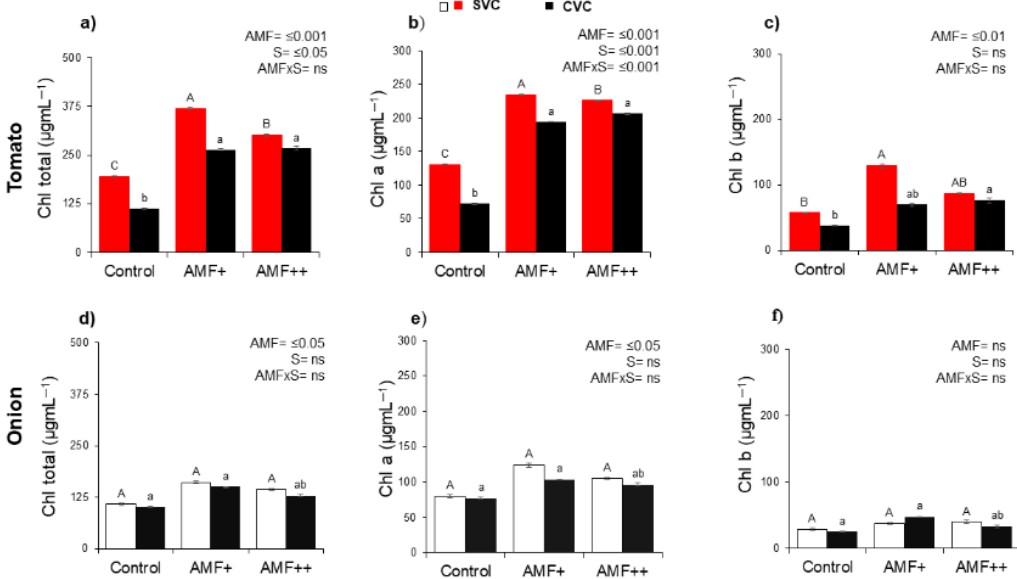

**Figure 3.** Effect of AMF inoculation on chlorophyll content in the tomato plant leaves (**a–c**) (Chl total, Chl a, Chl b) and onion (**d–f**) plant leaves grown on two different mixed substrates. Vertical bars show ± standard errors for mean (*n* = 3). Different letters (capital letters for SVC and lowercase letters for CVC) indicate significant differences according to the Tukey test (*p* < 0.05), and the *p*-value indicates a significance level based on two-way ANOVA.

**Table 4.** Effect of two commercial products of *R. irregularis* on macro- and micronutrients in tomato plant leaves grown on two mixed substrates.

|  | Treatments | N (%) | P (%) | Ct (%) | Fe (ppm) | Cu (ppm) | Zn (ppm) | Mn (ppm) |
|---|---|---|---|---|---|---|---|---|
| SVC | Control | 1.56 ± 0.07 b | 0.34 ± 0.08 b | 34 ± 0.67 a | 38 ± 1.21 c | 6 ± 0.45 c | 15 ± 2.07 b | 142 ± 0.20 b |
|  | AMF+ | 3.25 ± 0.07 a | 0.76 ± 0.05 a | 42 ± 0.33 a | 189 ± 0.34 a | 12 ± 0.26 b | 30 ± 0.35 a | 637 ± 1.01 a |
|  | AMF++ | 2.71 ± 0.04 a | 0.51 ± 0.08 a | 42 ± 0.13 a | 129 ± 0.90 b | 16 ± 0.54 a | 21 ± 0.36 a | 362 ± 0.78 b |
| CVC | Control | 1.93 ± 0.03 b | 0.47 ± 0.05 a | 32 ± 0.60 a | 56 ± 0.69 c | 8 ± 0.31 b | 11 ± 0.52 b | 93 ± 0.65 c |
|  | AMF+ | 3.06 ± 0.46 a | 0.68 ± 0.05 a | 39 ± 0.32 a | 163 ± 0.66 a | 19 ± 0.68 a | 35 ± 0.47 a | 124 ± 0.76 a |
|  | AMF++ | 1.95 ± 0.11 b | 0.46 ± 0.04 a | 40 ± 0.14 a | 77 ± 0.78 b | 19 ± 0.43 a | 21 ± 0.31 a | 167 ± 0.55 b |
|  | AMF | <0.001 | ns | ns | <0.001 | <0.001 | <0.001 | <0.001 |
|  | S | <0.05 | ns | <0.001 | <0.001 | ns | <0.001 | <0.001 |
|  | AMF × S | <0.01 | ns | ns | <0.001 | ns | <0.001 | <0.001 |

Means within a column followed by the same letter(s) are not significantly different (ns) at *p* > 0.05 based on the Tukey test. The *p*-value indicates a significance level based on a two-way ANOVA. AMF+, soil-multiplied *R. irregularis*; AMF++, in vitro *R. irregularis*; AMF, arbuscular mycorrhizal fungi; S, substrate, AMF × S, indicates interaction between AMF and substrate.

### 3.6. Photosynthetic Activities

Generally, AMF inoculation of the SVC and CVC growth substrates favored photosynthetic activity (Ci, *Pn*, *E*, and *gs*) in the tomato plants compared with the non-inoculated plants. In particular, AMF+ inoculation significantly increased the Ci (37%), *Pn* (51%), *E* (78%), and *gs* (48%) in the SVC growth substrate compared with the non-inoculated plants. However, AMF++ inoculation showed a less significant increase in Ci (25%), *Pn* (33%), *E* (63%), and *gs* (44%) in the CVC growth substrate (Figure 4).

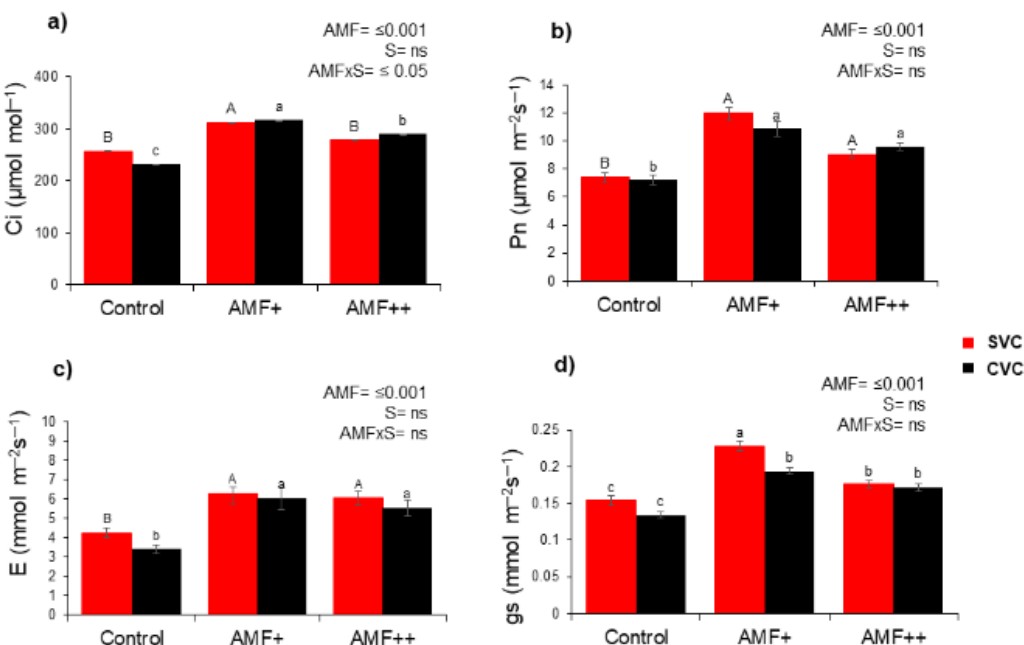

**Figure 4.** Effect of AMF inoculation in tomato: (**a**) $CO_2$ internal concentration (Ci), (**b**) net photosynthesis (*Pn*), (**c**) leaf transpiration rate (*E*), and (**d**) stomatal conductance (*gs*) grown in two different mixed substrates. Vertical bars show ± standard errors for mean (*n* = 3). Different letters (capital letters for SVC and lowercase letters for CVC) indicate significant differences according to the Tukey test (*p* < 0.05) and the *p*-value indicates a significance level based on two-way ANOVA.

## 4. Discussion

The level of AMF colonization was slightly higher in the SVC substrate than in the CVC substrate in both crops, and it was higher when using the AMF multiplied (AMF+) in soil. This result is in accordance with previous works showing that the mycorrhization rate varies with the fertility of the soil, especially with regards to the P content (Gamalero et al., 2002). In this work, the SVC substrate had a lower content of P (Table 1) than the CVC substrate, so the physical and chemical conditions in the SVC substrate could be related to a better level of AMF colonization, as stated in other studies [43,44]. In the substrate, the source of the inoculum caused a differential response, whereas the soil-multiplied inoculum showed greater results in terms of the growth of the plants due to its acclimatization to the specific substrate conditions, including pH and nutrient availability [45]. As a result, the soil-multiplied inoculum is better suited to the local environment, leading to improved colonization and the establishment of arbuscular mycorrhizal associations with plant roots when compared with in vitro AMF.

The variation in the beneficial effect of *R. irregularis* on the above- (morpho-agronomic) and belowground (root architecture) characteristics was well related to the level of mycorrhization. On one hand, it has been reported that root architecture can be changed in response to AMF inoculation in a suitable growth substrate [46], and such effects might be varied considerably with crop species, suggesting that the positive effect of AMF species with a suitable growth substrate is likely related to crop species [47]. In this study, AMF play a crucial role in promoting the root architecture of the tomato and onion plants, having a favorable effect on their growth patterns, branching, and overall structure (Figure 4). Consequently, the AMF *R. irregularis* (both in vitro and soil-multiplied) positively affected the tomato and onion root development by enhancing several root parameters in both mixed substrates compared with the control plants. The increases induced on several tomato and onion root parameters by *R. irregularis* are in agreement with those recorded by Smith and Read [15] in which AMF-inoculated tomato roots exhibited a higher number of lateral roots and roots hairs (74), along with other plants and AMF species [20,43,48,49]. According to our study, the influence *R. irregularis* on increasing the root system was shown, resulting

in a better exploration volume of substrate, increased absorption, and the facilitation of their transport and exchange within the root cells. Our results revealed that an enhanced root system and AMF+ inoculation in the SVC substrate had a favorable impact on the agronomical parameters in tomato and onion intercropping, whereas AMF++ also showed some promising effects. In fact, both AMF commercial products were able to colonize the tomato and onion plants, increasing the plant height and stem diameter. These results are in accordance with previous studies in which *R. irregularis* inoculation was applied to tomato [50–52] and onion [53] plants. On the other hand, the SVC substrate, with its superior draining properties, provided greater plant height and stem diameter, and may indirectly contribute to thicker stems [28,54].

In this study, despite the better root percentage shown by AMF *R. irregularis* colonization in onion plants (35–48%), only a significant increase in the chlorophyll content of the tomato plants was observed compared with the onion plants. The increase in chlorophyll content in the AMF-colonized tomato leaves can be attributed to the extended hyphal network of AMF in the soil, facilitating the absorption of nutrients [55], which are crucial for chlorophyll synthesis [56], improving photosynthetic capacity [57], and collectively contributing to improving plant growth and development. However, it is important to consider that the specific effects on chlorophyll content shown by *R. irregularis* may vary depending on factors such as the specific AMF species, plant genotype, environmental conditions, and experimental design [58]. In this sense, it has been reported that some plants have shown significant changes in chlorophyll content in response to AMF colonization, while others may not exhibit such pronounced effects [59]. Some previous studies have revealed that AMF may not have a significant direct impact on onion chlorophyll content [9], with the same response obtained in our study, so it could be possible that onion plants have a different level of interaction with AMF compared with tomato plants. The specific physiological and biochemical responses of onions to AMF may differ, potentially resulting in limited or no direct impact on chlorophyll content. It may be possible that AMF have a more significant impact on onion growth and nutrient uptake under nutrient-limiting or stressful conditions [60]. After obtaining these results regarding chlorophyll in onion plants, it was decided to further evaluate the effect of *R. irregularis* on leaf macro- and micronutrient status, and their impact on photosynthetic parameters in tomato plants.

It was observed that AMF colonization (29–44%) enhanced the root system architectural parameters of the tomato plant, contributed to capturing minerals [61], and improved the macronutrient and micronutrient status and overall nutritional health of the growing plants, as previous studies have reported [62–64]. In particular, in this research, an increase in macronutrients (N and P), and especially in micronutrients (Fe, Mn, Zn, and Cu), and even the root percentage, was observed as a result of AMF inoculation. It has been reported that AMF enhances the availability of micronutrients by increasing their solubility and uptake efficiency [65], producing organic acids and enzymes that can chelate and release micronutrients from the soil, making them more accessible to the plant roots. These elements are essential for various physiological processes, such as enzyme activities, chlorophyll synthesis, and photosynthetic activities.

Previous studies have also shown that a greater chlorophyll content represents higher rates of photosynthesis and carbon fixation, sustaining arbuscular plant symbiosis. In our study, we found that AMF inoculation favored these key physiological parameters in tomato leaves [66,67]. AMF colonization improves the carbon sink that stimulates the plant to increase photosynthesis to fulfill the carbon demand [68]. It has been reported that this increased sink strength of colonized roots helps with the faster removal of sugars from leaves, enabling higher photosynthetic rates [69,70]. This results in an increased concentration of sugars in inoculated plants compared with non-inoculated plants. The higher root carbon favors the extra metrical mycelium that can improve the nutrient status of plants. Research by Leventis et al. [71] demonstrated that AMF-inoculated tomato plants exhibit increased stomatal conductance compared with non-inoculated plants, indicating an enhanced gas-exchange capacity. In addition to influencing net photosynthesis, AMF

colonization can also affect the transpiration rate in tomato leaves, which plays an important role in water transport and overall plant physiology. On the other hand, these mixed substrates, when supplemented with an appropriate nutrient plan, can provide an aerated environment for roots, supporting essential physiological processes in plants [72].

## 5. Conclusions

The application of a commercial product, *Rhizophagus irregularis*, as soil-multiplied or an in vitro inoculum, colonized the roots of intercropped tomato and onion to positively stimulate plant growth and root architecture traits on SVC and CVC growth substrates. In particular, the main effects of AMF and the growth substrate were also reflected in the mineral and chlorophyll contents, improving the photosynthetic activity in the tomato plants. The main findings of this study show highly significant increases in micronutrients such as Fe, Cu, Mn, and Zn in the tomato leaves. Furthermore, AMF positively stimulated the root structure to facilitate water and mineral uptake from the substrate for better plant development. However, both crops grew better in the sand-mixed growth substrate compared with the compost-mixed substrate. Therefore, it can be speculated that the sand-mixed growth substrate was the main driver in producing the beneficial effects of AMF. Moreover, the potential combination of AMF use along with cheap and environmentally friendly substrates can be used in commercial agriculture. The application of AMF with suitable growth substrates could substantially promote sustainability efforts. Consequently, greater collaboration between farmers, industrial sectors, researchers, and governmental entities is required to improve production systems and AMF quality to create and implement improved and environmentally friendly agricultural practices.

**Author Contributions:** M.S. conceived and conceptualized the idea, performed the literature review, conducted the experiments, collected the data, performed statistical analysis, and prepared the first draft; C.N.-L., J.C.-S. and D.R.G.-E. provided technical expertise to strengthen the basic idea and acquired funds; M.K. edited and reviewed each draft, and provided co-funding; S.Y. and A.M.-U.; proofread and provided intellectual guidance regarding statistical analysis. All authors read the first draft, helped revise, and approved the article. All authors have read and agreed to the published version of the manuscript.

**Funding:** This research was co-funded by the Centro Universitario de Ciencias Biológicas y Agropecuarias (PROSNI, PROINPEP-BEMARENA Y P3E 2020-2022), Universidad de Guadalajara and CONAHCYT, Mexico for a scholarship (CVU:962791) to the senior author.

**Data Availability Statement:** Not applicable.

**Acknowledgments:** The authors are thankful to the anonymous reviewers for providing valuable comments.

**Conflicts of Interest:** The authors declare no conflict of interest.

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
