# Peer review of "Arbuscular Mycorrhizal Fungi as a Plant Growth Stimulant in a Tomato and Onion Intercropping System"

_agronomy, doi:10.3390/agronomy13082003_

Round 1

Reviewer 1 Report

The authors studied arbuscular Mycorrhizal Fungi (specifically Rhizophagus irregularis) as plant growth stimulant in tomato and onion intercropping system. The study is interesting and it is worthwhile to publish. However, tehre are few details about the english usage to take into account before its acceptation. I am attaching the pdf with some annotations.

The authors studied arbuscular Mycorrhizal Fungi (specifically Rhizophagus irregularis) as plant growth stimulant in tomato and onion intercropping system. The study is interesting and it is worthwhile to publish. However, tehre are few details about the english usage to take into account before its acceptation. I am attaching the pdf with some annotations.

Author Response

Comments and Suggestions for Authors: Reviewer 1

Comment 1: The authors studied arbuscular Mycorrhizal Fungi (specifically Rhizophagus irregularis) as a plant growth stimulant in tomato and onion intercropping system. The study is interesting and it is worthwhile to publish. However, there are few details about English usage to take into account before its acceptance. I am attaching the pdf with some annotations.

Response: Thank you for your valuable feedback and for acknowledging the importance of our study on the use of arbuscular Mycorrhizal Fungi Rhizophagus irregularis as a plant growth stimulant in a tomato and onion intercropping system. We appreciate your thorough evaluation of the manuscript. We have taken your comments regarding the English language into serious consideration. In response, the manuscript was submitted to the MDPI services for English revision.

We would like to express our gratitude for your guidance and suggestions, which have significantly contributed to improving the manuscript. We have carefully reviewed the attached PDF with your annotations and have made the necessary revisions accordingly.

Reviewer 2 Report

The current manuscript entitled “Arbuscular Mycorrhizal Fungi as plant growth stimulant in tomato and onion intercropping system” was conducted to examine the effects of two commercially available arbuscular mycorrhizal fungi (AMF) products of Rhizophagus irregularis on the growth of tomato and onion plants using two different mixed growth substrates (river sand and compost) in a greenhouse setting. The effect of AMF inoculation on plant growth, chlorophyll content, photosynthetic activity, leaf mineral contents, root morphology, and root colonization percentage were estimated.

Comments:

The materials and methods section should provide a detailed description of the sterilization method employed for the growth substrates, which consisted of volcanic rock and coconut fiber mixed with either river sand (SVC) or compost (CVC).

To assess the impact of AMF inoculation on plant growth, it is crucial to determine the fresh and dry weights of shoots and roots as significant parameters.

The examination of AMF colonization percentages in control (non-inoculated) plants is essential. If the AMF colonization of control plants was found to be 0%, the authors should provide commentary on these results.

Regarding the statistical analysis to compare the effects of AMF inoculation within the two different mixed growth substrates (river sand and compost), it should be clearly outlined in the materials and methods section. Additionally, the results section should present the statistical analysis within the tables and figures.

In the discussion section, it is important to explain the relationship between AMF colonization percentage and leaf mineral concentration. Several clarifications pertaining to the AMF treatments and estimated parameters should be provided in this section.

Rhizophagus irregularis should be italic all over the manuscript.

Minor editing of English language required

Author Response

Comments and Suggestions for Authors: Reviewer 2

Comment 1: The materials and methods section should provide a detailed description of the sterilization method employed for the growth substrates, which consisted of volcanic rock and coconut fibre mixed with either river sand (SVC) or compost (CVC).

Response: Thank you for your valuable suggestion regarding the materials and methods section of our manuscript.

We have carefully considered your feedback and have made the necessary revisions to include a detailed description of the sterilization method employed.

Comment 2: To assess the impact of AMF inoculation on plant growth, it is crucial to determine the fresh and dry weights of shoots and roots as significant parameters.

Response: Thank you for your valuable comment regarding the assessment of the impact of AMF inoculation on plant growth. We completely agree with your suggestion that determining the fresh and dry weights of shoots and roots is crucial in evaluating the effects of AMF on plant growth. However, we based our research objectives on the idea of measuring biometric parameters in non-destructive manners such as plant height, shoot diameter, leaf number, the state of flowering, nutritional status of plant shoots, transpiration rates, photosynthesis, chlorophyll concentration or intracellular CO2 concentration in leaves which are already determined mostly non-destructively according to Füzy et al., 2019 (https://link.springer.com/article/10.1007/s11738-019-2842-9).

The decision was also based on the following study ideas that morphological traits such as plant height and leaf characteristics should be considered in the evaluation as they are important for plant growth and development assessments, including physiological parameters, therefore these data may provide early information related to posterior crop performance (Dubberstein et al., 2020) https://www.geneticsmr.com/articles/biometric-traits-tool-identification-and-breeding-coffea-canephora-genotypes.

Comment 3: The authors should provide commentary on these results.

Response: We have added the suggested explanation.

Comment 4: Regarding the statistical analysis to compare the effects of AMF inoculation within the two different mixed growth substrates (river sand and compost), it should be clearly outlined in the materials and methods section. Additionally, the results section should present the statistical analysis within the tables and figures.

Response: Thank you for your valuable feedback regarding the statistical analysis in our manuscript.

In the materials and methods section, we tried to provide a clear and detailed outline of the statistical analysis used to compare the effects of AMF inoculation within the two different mixed-growth substrates (river sand and compost). This information will help readers understand the methodology employed for data analysis and interpretation. Furthermore, in the results section, we have presented the statistical analysis within the tables and figures, making the results more transparent and easily interpretable. This inclusion enhances the clarity of the findings and supports the robustness of our conclusions.

Comment 5: In the discussion section, it is important to explain the relationship between AMF colonization percentage and leaf mineral concentration. Several clarifications pertaining to the AMF treatments and estimated parameters should be provided in this section.

Response: Thank you for your insightful comments regarding the discussion section of our manuscript. We acknowledge the importance of explaining the relationship between AMF colonization percentage and leaf mineral concentration, as well as providing further clarifications on the AMF treatments and estimated parameters.

We have carefully considered your suggestions and have made the necessary additions and clarifications in the discussion section. Specifically, we have included a detailed explanation of the relationship between AMF colonization percentage and leaf mineral concentration, shedding light on the underlying mechanisms and potential implications.

Furthermore, we have provided additional information on the AMF treatments and estimated parameters, ensuring a more comprehensive understanding of our experimental setup and results interpretation.

Comment 6: Rhizophagus irregularis should be italic all over the manuscript.

Response: Thanks for the suggestions. We have made the necessary changes

Comment 7: Comments on the Quality of the English Language: Minor editing of the English language is required.

Response: We have taken your comments regarding the English language into serious consideration. In response, the manuscript was submitted to the MDPI services for English revision.

Reviewer 3 Report

The whole article is relatively simple. The effect of adding AMF on the physiological and biochemical processes of crops should be investigated thoroughly. In addition, the data from one trial was not robust enough, and at least three different times required.

 Introduction

In the introduction section, you need to specify What is your aim and how to conduct your research topic.

 Line 181: length (cm), diameter (cm) ?

 In the Materials and Methods section, more detail descriptions related to time of the sample obtained are needed,

 In the Result section, the presence of “significant” word requires a p-value. Please check the related place in the Result section,

 To many sections in the Results section, please merge related contents.

 Line 317-330: Delete it or replace it to another place. It is not suitable to begin the Discussion section like this.

Improve!

Author Response

Comments and Suggestions for Authors: Reviewer 3

Comment 1: The whole article is relatively simple. The effect of adding AMF on the physiological and biochemical processes of crops should be investigated thoroughly. In addition, the data from one trial was not robust enough, and at least three different times were required.

Response: Thank you for your valuable comments on our article. We appreciate your feedback and suggestions for further improvement. While we acknowledge that the experimental design of our study may appear relatively simple, we believe that its simplicity is balanced by its novelty and the valuable insights it provides into crop production systems for sustainable management. Regarding the data from one trial, we understand your concern. However, we would like to emphasize that we conducted the experiment over a period of 65 days to ensure the significant effects of our treatment on the plants. We believe that the data collected during this duration provide sufficient evidence to support our conclusions. In general, several published works address only one trial due to the vast quantity of work for evaluating several parameters. Indeed, research work with different times over different seasons would be desirable, once this model has been approved for publication it could be escalated to a more complex design through several trials in future.

Comment 2: In the introduction section, you need to specify what is your aim is and how to conduct your research topic.

Response: Thanks for your comment. We have revised the objective of our study and now it is more understandable and clear.

Comment 3: Line 181: length (cm), diameter (cm)?

Response: We have revised the measurement units.

Comment 4: In the Materials and Methods section, more detailed descriptions related to the time of the sample obtained are needed,

Response: Thanks for pointing out the mistake; we have included the harvesting timetable in the method section.

Comment 5: In the Result section, the presence of a “significant” word requires a p-value. Please check the related place in the Result section,

Response: we have presented the statistical analysis within the tables and figures, making the results more transparent and easily interpretable. This inclusion enhances the clarity of the findings and supports the robustness of our conclusions.

Comment 6: Too many sections in the Results section please merge related contents.

Response: Thank you for your valuable feedback on our manuscript. We appreciate your suggestion to merge related content in the results section. We have carefully considered your comment and made the necessary revisions to merge related sections. By doing so, we aimed to improve the clarity and coherence of the Results section, ensuring that the information is presented in a logical and reader-friendly manner.

Comment 7: Line 317-330: Delete it or replace it to another place. It is not suitable to begin the Discussion section like this.

Response: We have deleted it from the manuscript.

Comment 8: Comments on the Quality of English Language: Improve!

Response: We have taken your comments regarding the English language into serious consideration. In response, the manuscript was submitted to the MDPI services for English revision.

Round 2

Reviewer 2 Report

Thanks for the authors for their efforts to improve the current version. Authors addressed most of my comments and the current version of the manuscript was improved and acceptable for publication.